# Current *Helicobacter pylori* Diagnostics

**DOI:** 10.3390/diagnostics11081458

**Published:** 2021-08-12

**Authors:** Dmitry S. Bordin, Irina N. Voynovan, Dmitrii N. Andreev, Igor V. Maev

**Affiliations:** 1A.S. Loginov Moscow Clinical Scientific Center, 111123 Moscow, Russia; i.voynovan@mknc.ru; 2A.I. Yevdokimov Moscow State University of Medicine and Dentistry, 127473 Moscow, Russia; dna-mit8@mail.ru (D.N.A.); igormaev@rambler.ru (I.V.M.); 3Tver State Medical University, 170100 Tver, Russia

**Keywords:** *Helicobacter pylori* (*H. pylori*), diagnosis, endoscopy, artificial intelligence, histology, molecular methods, serology, stool antigen test, urea breath test

## Abstract

The high prevalence of *Helicobacter pylori* and the variety of gastroduodenal diseases caused by this pathogen necessitate the use of only accurate methods both for the primary diagnosis and for monitoring the eradication effectiveness. There is a broad spectrum of diagnostic methods available for detecting *H. pylori*. All methods can be classified as invasive or non-invasive. The need for upper endoscopy, different clinical circumstances, sensitivity and specificity, and accessibility defines the method chosen. This article reviews the advantages and disadvantages of the current options and novel developments in diagnostic tests for *H. pylori* detection. The progress in endoscopic modalities has made it possible not only to diagnose precancerous lesions and early gastric cancer but also to predict *H. pylori* infection in real time. The contribution of novel endoscopic evaluation technologies in the diagnosis of *H. pylori* such as visual endoscopy using blue laser imaging (BLI), linked color imaging (LCI), and magnifying endoscopy is discussed. Recent studies have demonstrated the capability of artificial intelligence to predict *H. pylori* status based on endoscopic images. Non-invasive diagnostic tests such as the urea breathing test and stool antigen test are recommended for primary diagnosis of *H. pylori* infection. Serology can be used for initial screening and epidemiological studies. The histology showed its value in detecting *H. pylori* and provided more information about the degree of gastric mucosa inflammation and precancerous lesions. Molecular methods are mainly used in detecting antibiotic resistance of *H. pylori*. Cultures from gastric biopsies are the gold standard and recommended for antibiotic susceptibility tests.

## 1. Introduction

*Helicobacter pylori* (*H. pylori*) is one of the most common human pathogens and a leading etiological factor for various gastroduodenal diseases, including chronic gastritis, peptic ulcers, gastric adenocarcinoma, and MALT lymphoma [1,2]. According to the latest systematic review with meta-analysis, 44.3% (95% CI: 40.9–47.7) of the global population are infected with this microorganism [3]. Timely diagnosis and subsequent eradication of *H. pylori* in adults allows one to resolve inflammatory changes in the gastric mucosa and prevent the development of precancerous conditions (atrophic gastritis and intestinal metaplasia) [4,5,6].

There are several diagnostic methods for detecting *H. pylori* infections. All methods can be broadly classified as invasive or non-invasive (Table 1). Invasive methods require upper endoscopy and analysis of the gastric biopsy. Preference should be given to non-invasive diagnostic methods. If the patient requires upper endoscopy, a histological analysis, rapid urease testing, molecular methods, or culture can be performed to diagnose the *H. pylori* infection [7,8]. The main limitation of these methods is their invasiveness and the ability to analyze only a small part of the gastric mucosa. Table 1 shows the general characteristics of the diagnostic methods for *H. pylori*, their applications in clinical practices, as well as the choice of diagnostic tests in different clinical conditions. Non-invasive tests include immunological methods (serology, stool antigen test), the 13 C-urea breath test (UBT), and molecular methods, i.e., a PCR study with determination of *H. pylori* DNA in feces (PCR from stool) [7].

Any of the tests can be used for the primary diagnosis of *H. pylori*. The urease breath test is a “gold standard” in the diagnosis of *H. pylori* infection [7,8,9,10].

Modern non-invasive tests provide high reliability in *H. pylori* detection due to their high sensitivity and specificity. However, all of these methods have limitations. The choice of a particular test will depend on its sensitivity, specificity, and the clinical circumstances [11].

## 2. Invasive Methods for *H. pylori* Diagnostics

### 2.1. Endoscopic Imaging

Upper endoscopy is of particular importance in the diagnosis of *H. pylori* gastritis since *H. pylori* infection is strongly associated with gastric carcinogenesis. However, studies have shown that conventional image-enhanced endoscopy (IEE) with white light imaging (WLI) does not allow for the diagnosis of a wide range of inflammations of the gastric mucosa [12].

There is growing interest in improving the visualization of pathological changes in the gastric mucosa and in detecting *H. pylori* infections in real time during an upper endoscopy. *H. pylori* imaging in real time could reduce the cost of diagnosis and treatment.

Narrow band imaging is a new method of visual endoscopy based on the use of a laser light source and has opened up new possibilities for the diagnosis of not only precancerous changes in the gastric mucous but also of *H. pylori* infection.

A new IEE system that has two laser light sources offers four observation modes of white light imaging (WLI), blue laser imaging (BLI), BLI-bright, and linked color imaging (LCI). This is a new method of visual endoscopy developed in Tokyo called LASEREO (FUJIFILM Co., Tokyo, Japan) [13].

LCI and blue laser imaging (BLI) provide brighter endoscopic views and facilitate the diagnosis of inflammation and atrophy of the mucosal surfaces, allowing for the diagnosis of early gastric cancer. BLI improves the detection rate of early gastric cancer in comparison with that of white light imaging (93% vs. 50%, respectively, *p* < 0.001).

One study evaluated diffuse redness of the fundic mucosa, an endoscopic feature that could be correlated with *H. pylori* infection. The sensitivity and specificity for the diagnosis of *H. pylori* using LCI was higher (85.8; 93.3 and 78.3%, respectively) compared to that of WLI (74.2; 81.7 and 66.7%, respectively) [12].

Magnifying endoscopy (ME) is another IEE tool that allows for the predicting of *H. pylori* by the microvascular architecture of the gastric mucosa. A meta-analysis was carried out to assess the diagnostic performance of ME to predict *H. pylori* infection. One endoscopic diagnosis criterion of *H. pylori* was “pit plus vascular pattern”. The meta-analysis showed a high level of diagnostic accuracy of ME in predicting *H. pylori* infection [12]. ME accurately predicted *H. pylori* infection in both the white-light and chromoendoscopy modes [12]. However, ME requires specialized training in the interpretation of the images; therefore, is not widely used in everyday practice.

#### Artificial Intelligence

With the progress in computer technologies, artificial intelligence (AI) technologies have recently been applied in medicine to improve the quality of the diagnoses of diseases, to make an accurate diagnosis, and to predict disease progression and treatment planning [14]. Artificial intelligence, or neural networks known as “deep learning”, is based on training computers on datasets containing a large number of images with their corresponding labels. The neural network then uses these learned functions to classify a given image [13].

Previous studies have demonstrated the capability of artificial intelligence in the prediction of *H. pylori* infection status for diagnosing gastritis. AI was efficiently created with IEE, BLI, and LCI. The studies evaluated the diagnostic accuracy of WLI and IEE for *H. pylori* gastritis, which found it to be 83.8% for *H. pylori* infection using WLI with the magnifying function. One pilot study showed that artificial intelligence based on BLI and LCI demonstrated an excellent ability to diagnose *H. pylori*. Sensitivity for BLI-bright and LCI was 96.7% and 10% superior to that using WLI [13].

A systematic review and meta-analysis were performed for assessing artificial intelligence in the forecasting of *H. pylori* infection, presenting diagnostic performance. The accuracy of the AI algorithm reached 82% for the discrimination between images of no infection and post-eradication images [15].

Artificial intelligence offers promising diagnostic performance using endoscopic imaging. It can help identify neoplastic or non-neoplastic lesions of the gastric mucosa and gastric cancer at an early stage and detect *H. pylori* in real time [15]. Soon, AI-assisted endoscopy will be feasible in clinical practice.

### 2.2. Histology

Histology is still one of the most commonly used diagnostic methods. This method allows for direct visualization of *H. pylori* and can be recommended for primary diagnosis if upper endoscopy is required. In addition to routine hematoxylin and eosin, various selective stains are used to detect *H. pylori* such as Warthin–Starry, Hp silver stain, Dieterle, Giemsa, Gimenez, acridine orange, McMullen, and immunostaining. Giemsa staining has become the most used method worldwide for the detection of *H. pylori* due to its low cost, ease of use, sensitivity, and reproducibility. It should be borne in mind that *H. pylori* can be detected only on sufficiently thin and well-stained sections [16].

It is recommended to take at least two biopsies to identify *H. pylori*; the best option is two biopsies from the antrum and one from the corpus. Biopsy from the corpus is especially valuable for yielding positive results if the patient has been taking PPI for a long time when *H. pylori* is translocated from the antrum to the corpus [16] and with a background of atrophic gastritis.

Moreover, in the area of intestinal metaplasia, *H. pylori* in most cases is not detected either with conventional or various selective stains. The disappearance of *H. pylori* correlates with the development of intestinal metaplasia and a decrease in gastric secretion [17]. The accuracy of the method can be affected by low bacterial density, for example, from taking PPIs for a long time or an uneven distribution of *H. pylori* on the surface of the gastric mucosa [16].

The Maastricht V Consensus Report recommends patients to stop taking antibiotics and bismuth 4 weeks before the test and PPIs 2 weeks before testing [10].

The specificity of the histological method can reach 100%, and the sensitivity can reach 91–93% [18]. Some studies show that the sensitivity of these tests’ ranges from 50% to 95% and depends on the quality, location, size, and frequency of the biopsy and the applied staining varieties [8]. Hematoxylin and eosin staining of biopsies has very poor sensitivity (66%) and suboptimal specificity (88%). The histological sensitivity decreases to 70% in patients with peptic ulcer bleeding; however, it remains a quite reliable test compared with the rapid urease test or culture, regardless of the presence of the bleeding [16].

Additional staining in gastric biopsies was investigated, such as using cresyl violet or immunohistochemistry for *H. pylori* detection [19,20,21]. Benoit A et al. [20] reported that it is not necessary to use this method to detect a *H. pylori* infection since conventional selective stains show good diagnostic accuracy. Immunohistochemistry can be used in cases of low bacterial density, atrophic gastritis with extensive intestinal metaplasia, and chronic active gastritis without *H. pylori* identification by standard staining. Immunohistochemistry is more specific; however, it is more expensive and not available in all laboratories.

A novel method using a γ-glutamyl transpeptidase (GGT) activatable fluorescent probe was proposed this year. The γ-glutamyl hydroxy methyl rhodamine green probe reacts with GGT and immediately produces fluorescence. The method allows for the quantification of the GGT activity of *H. pylori* on gastric biopsies within 15 min. However, the sensitivity is still limited (75–82%) [22].

Despite the high specificity and sensitivity, the histology has a higher cost and longer processing time, requires an upper endoscopy to obtain gastric biopsy samples, depends on the skills of the operator, and is not suitable for assessing the effectiveness of eradication since endoscopy is necessary.

The main advantage of histology is the ability to assess the condition of the gastric mucosa and diagnose precancerous lesions. The degree and stage of chronic gastritis, risk of carcinogenesis, and assessment according to the modern classification of chronic gastritis (OLGA—Operative Link for Gastritis Assessment and OLGIM) allow for the assessment of the prognosis of the disease [23]. The updated Sydney System recommends taking five biopsy specimens from different sites for the assessment of the degree and stage of *H. pylori* gastritis status. According to this system, two biopsies are taken from the antrum (from the lesser and greater curvature, both within 2–3 cm from the pylorus), two from the corpus (the lesser curvature about 4 cm proximal to the angulus; the middle portion of the greater curvature, approximately 8 cm from the cardia), and one from the incisura angularis [24].

Atrophic gastritis (AG) and intestinal metaplasia (IM) are considered precancerous lesions of the stomach. Studies have shown that with AG and IM, the sensitivity of histology for detecting *H. pylori* infection decreases to 30–55%, while the corpus lesser curvature side showed 80% sensitivity, and the corpus greater curvature side showed 95–100% sensitivity. Thus, the appropriate biopsy site for detecting *H. pylori* infection in AG and IM patients as well as in patients with gastric cancer is the corpus, especially the corpus greater curvature side [16].

### 2.3. The Rapid Urease Test

The rapid urease test (RUT) is based on detecting the activity of the *H. pylori* urease enzyme, which splits the urea test reagent to form ammonia. Ammonia increases pH, which is detected by the phenol red indicator [7,25,26].

The RUT is a low cost, rapid, and generally highly specific assay.

The Maastricht V Consensus Report allows for the use of RUT for primary diagnosis, and a positive test result allows for the prescription of eradication, but it does not recommend a rapid urease test to assess eradication after treatment due to its lack of sensitivity and high false-negative rate [10]. Therefore, a negative rapid urease test should not be used to exclude *H. pylori*, which should also be taken into account in the initial diagnosis.

Commercially available RUT*s* (e.g., HpFast, GI-supply, Camp Hill, Pennsylvania; CLOTest, Delta West, Bentley, Western Australia; HpOne, GI Supply, Camp Hill, PA) have reported specificities from 95% to 100%, but their sensitivity is moderate (85% to 95%) [7,26]. However, the sensitivity of the test increases if we take biopsies from both the corpus and antrum [19].

RUT has limited sensitivity and can give false-negative results, for example, if less than 10^4^ bacterial cells are present in the gastric biopsy or if a biopsy is taken from areas of atrophy and metaplasia of the gastric mucosa. It is necessary to exclude the use of antibiotics and bismuth for 4 weeks and PPIs for 2 weeks before the test [7,8].

In some instances, RUT may lead to false-positive test results due to the presence of other urease-producing bacteria such as *Staphylococcus capitis* subsp. ureolyticus, *Streptococcus salivarius,* and *Proteus mirabilis* in the stomach [8]. Bleeding peptic ulcers reduce the sensitivity of RUT by up to 70%.

False-negative test results are more common than false-positive test results, so a negative result cannot be used to exclude a diagnosis of *H. pylori*. Thus, a positive RUT result indicates the presence of *H. pylori* and makes it possible to prescribe treatment, but a negative result does not allow excluding *H. pylori*; therefore, it is recommended to confirm the diagnosis with an additional method [10].

### 2.4. Culture

The greatest information about *H. pylori* can be obtained in isolation cultivations of *H. pylori* from gastric biopsy specimens. The cultivations allows not only for the isolation of a pure culture of *H. pylori* and its identification, but also the study of the morphological, biochemical, and biological properties of the pathogen and the pathogenicity factors of *H. pylori*. The bacteriological method of research makes it possible to the determine antibiotic resistance in *H. pylori* and carry out dynamic monitoring of it [7].

Bacteriological examination is a very laborious method; it requires taking at least two biopsies from the stomach. It is necessary to strictly follow the rules of transporting biopsy material for culture in order to keep this microorganism in a viable state. It is advisable to sow the material on the day it arrives at the laboratory. The incubation of crops is carried out under microaerophilic conditions with an oxygen content of ≤5%. Later, the cultures are identified, and their morphological and tinctorial properties and sensitivity to antibiotics (e.g., amoxicillin, clarithromycin, and metronidazole) are determined [7].

The specificity of the method is 100% when performed under optimal conditions; the sensitivity is 76–90% [16], and according to other data it is 50–90% [17].

As with any diagnostic method, the bacteriological research method not only has advantages, but also has disadvantages, which often limit the widespread use of this method in clinical practice. Most importantly, the shortcomings include the need for special laboratory equipment and reagents, special nutrient media, and trained specialists. This is all associated with high material costs.

False-negative results arise from non-adherence or inaccurate adherence to the test method, such as poor sample quality, delayed transport, exposure to an aerobic environment, or an inexperienced microbiologist [7].

Patient factors such as low bacterial load; bleeding from the upper gastrointestinal tract; alcohol consumption; or taking PPIs, bismuth preparations, H_2_RA, and antibiotics have an adverse effect on obtaining a culture of *H. pylori* [7].

PPIs, H_2_RA, bismuth, and antibiotics should be stopped 4 weeks before the culture method. To avoid negative results due to the uneven distribution of *H. pylori* in the stomach and to increase the sensitivity and specificity of the method in the diagnosis of *H. pylori*, it is necessary to take several biopsies from the gastric mucosa: two from the antrum and two from the body of the stomach. Some authors believe that in order to increase the sensitivity and specificity of the bacteriological method, taking biopsies for cultivation should be carried out 3 months after patients cease taking PPIs, antibiotics, and bismuth [27].

Although the culture is very laborious and requires special conditions for implementation, it is very valuable in clinical practice. The Maastricht V Consensus Report recommends culture and antibiotic-susceptibility testing in geographical areas where primary resistance to clarithromycin is higher than 20%. This method is recommended after failure of second-line treatments, when the further choice of antibiotics is determined by the sensitivity of *H. pylori* to them [10].

## 3. Non-Invasive Methods for *H. pylori* Diagnostics

### 3.1. Urea Breath Test

13C-UBT is a non-invasive method for the diagnosis of *H. pylori* based on a simple principle: patients ingest urea labeled with 13C or 14C, and *H. pylori* produces urease—an enzyme that splits urea into ammonia and 13C-labeled carbon dioxide; then, 13C carbon dioxide is absorbed into the bloodstream, enters the lungs, and is excreted with the exhaled air [7].

Urea is usually given to the patient with a citrus juice (lemon, orange) to delay gastric emptying and increase contact time with the mucosa.

Before taking the test solution, the exhaled air is collected in a sealed bag 30 min after the solution has been drunk. The collected air samples are analyzed on a mass spectrometer or by infrared spectroscopy, which is technically simpler and also cheaper than using a mass spectrometer. Infrared spectroscopy determines the 13C/12C isotopic ratio. The increase in labeled CO2 is expressed as delta over baseline (DOB). The DOB value is positively correlated with the *H. pylori* bacterial load.

Thus, from the appearance of 13C in the exhaled air, we can determine with high accuracy whether the patient is infected with *H. pylori*, and from the value of the 13C/12C ratio, we can estimate the degree of infection. The 13C urea breath test is similar to the 14C urea breath test except that 13C is a non-radioactive isotope.

The standard urea breath test uses 75 mg of 13C. The sensitivity of 13C-UBT is 96–100%; the specificity is 93–100% [7,28].

One study found that testing time could be shortened to 15 min (the BREATH QUALITY UBT) without affecting the accuracy of the method [29].

The conducted meta-analysis showed the high accuracy of the test in children of any age. In children >6 years, sensitivity and specificity were 96.6% and 97.7%, respectively; in children ≤6 years, they were 95% and 93.5%, respectively [30].

Recently, a new UBT technique has been proposed, which uses a 13C-urea tablet formulation. This technique allows for air sampling with high accuracy within 10 min after taking the pill. In addition, the tablet form has the advantage of preventing the formulation from interacting with the urease-producing bacteria in the oropharynx, which can cause false-positive results [8,31].

False-positive results are rare, but they can be observed after endoscopy with a biopsy immediately before the test in patients who underwent gastric resection and also those with a significant decrease in gastric secretion. False-positive tests most often cause hydrolysis of urea by bacteria in the mouth or bacteria containing urease in the stomach [31]. This is especially likely in the presence of achlorhydria or hypochlorhydria. A small number of false-negative results may be associated with a violation of the method of taking and storing samples of exhaled air or physical activity on the eve of and during the test. As with most other tests, a reliable UBT result can be obtained after a 2-week discontinuation of PPIs and no earlier than 4 weeks after stopping antibiotics and bismuth [7,10].

### 3.2. Stool Antigen Test

The stool antigen test (SAT) is based on the direct identification of the bacterium antigen in stools. There are two types of SATs used for *H. pylori* detection: enzyme immunoassay (EIA) and speedy in-office tests—immunochromatography assay (ICA)-based methods, using either polyclonal antibodies or monoclonal antibodies. EIA provides more reliable results than does ICA. Monoclonal antibodies-based tests are more accurate than are polyclonal antibodies and give useful reports [32].

SAT is recommended both for the primary diagnosis of *H. pylori* infection and for the monitoring of therapy effectiveness. This test is noninvasive, quick, low cost, and easy to use. The test has a good sensitivity of 95.5% and a specificity of 97.6% (LIAISON^®^ Meridian) [7,22].

The test requires a small amount of feces, and it is possible to collect a sample at home and send it to a laboratory at a suitable time. Stool samples can be frozen at −20 °C and stored for a long time. It is important to remember that the sensitivity of the test drops to 69% if the sample is kept at room temperature for 48–72 h. It is not recommended to perform the test during diarrhea or on watery stools [33].

SAT must be performed not earlier than four weeks after last intake of antibiotics and bismuth or two weeks after the last intake of PPI. To evaluate the eradication efficiency, the test must be used 30 or more days after the completion of eradication [33]. Uneven distribution of antigen in feces, destruction of antigen in constipation, ongoing bleeding of the gastrointestinal tract, and low bacterial load in the stomach are the reasons for false-negative results [33,34].

Stool monoclonal antigen is a convenient and effective test for the diagnosis of *H. pylori* in children [35].

### 3.3. Serology

The colonization of *H. pylori* induces a systemic immune response. Antibodies to *H. pylori* appear in the blood 3–4 weeks after infection. These antibodies can be determined by one of three methods: the enzyme-linked immunosorbent assay (ELISA) test, latex agglutination tests, and Western blotting. Of these, ELISA is the most commonly used method [36]. This method is based on the detection of specific circulating antibodies: IgG, IgA, and IgM. *H. pylori* is a chronic infection; therefore, only a validated IgG test should be used [10].

Serologic tests are widely available to diagnose *H. pylori*; they are non-invasive, rapid, do not require any special equipment, and can be used in screening populations.

However, serology may be positive due to the presence of an active infection at the time of the test, a previous infection, or because of non-specific cross-reacting antibodies [7,8].

Immunoglobulins (antibodies) against antigens appear due to the presence of active infection, previous infection, or because of non-specific cross-reacting antibodies [36]. Thus, a serological test can be used for primary diagnosis of *H. pylori* or another test confirmation. Quantitative antibodies levels do not decline significantly for a long time after successful eradication; therefore, serological testing should not be used for therapeutic follow-up. Furthermore, false-positive serologic tests are common in a population with a low prevalence (<40%) of *H. pylori* as the positive predictive value of serology depends on the prevalence of *H. pylori* infection in the considered area [33]. In such populations, it is not recommended to use serology, and in case of positivity of a serological test for *H. pylori*, it is necessary to confirm the test with a more reliable test, e.g., histological tests, culture of biopsy sample, the urea breath test, or the stool antigen test.

Serology is not affected by recent use of proton-pump inhibitors, antibiotics, or bismuth preparations, gastrointestinal bleeding, or atrophy of the gastric mucosa [10].

The specificity and sensitivity of serological testing varies. One meta-analysis showed that the sensitivity and specificity of the test were 85% and 79%, respectively. Another study demonstrated sensitivity ranging from 76% to 84% and specificities from 79% to 90% [33].

Several studies have shown that the levels of anti-*H. pylori* IgG were associated positively with the grade of histological gastritis, mucosal bacterial density, and levels of serum biomarkers for stomach function, including PGI, PGII, PGI/II ratio, and gastrin-17. Other studies found no associations; thus, the results are conflicting [37].

## 4. Molecular Invasive and Non-Invasive Methods for *H. pylori*

Molecular diagnostic methods are based on the amplification of nucleic acid using a conventional polymerase chain reaction (PCR) or PCR in real time (RT-PCR). Genetic material (DNA) of *H. pylori* can be detected in gastric biopsy, saliva, feces, or dental samples. PCR can be considered as either an invasive or non-invasive method for detecting *H. pylori* depending on the applied material. It demonstrates up to 95% sensitivity and 95% specificity [38]. Molecular methods are more expensive than other methods, and the laboratory must have appropriate equipment and experience. PCR allows for the detection of specific mutations leading to antibiotic resistance and bacterial virulence factors such as CagA and VacA.

There are a number of molecular assays commercially available for *H. pylori* and clarithromycin-resistance detection. Several studies have found different sensitivities and specificities of the method depending on the DNA extraction method and the PCR assay used. The *H. pylori* Taqman^®^ real-time PCR assay in stool specimens shows a high sensitivity of 93.8%. The ClariRes assay shows a low sensitivity (ranging from 63% to 84%) for *H. pylori* detection in stool specimens when compared to those of the stool antigen test and *H. pylori* culture from gastric biopsy specimens [26].

One of the new approaches to diagnosing *H. pylori* is next-generation sequencing (NGS) by sequencing *H. pylori* DNA directly from formalin-fixed paraffin-embedded (FFPE) gastric biopsy specimens. NGS reveals mutations in genes that lead to resistance to antibiotics (clarithromycin, levofloxacin, and tetracycline) and their correlation with phenotypic drug resistance. Using NGS, mutations in the gyrA, 23S rRNA, and 16S rRNA genes were identified and analyzed [22]. The sensitivity of the method is 95%. The study showed the possibility of using NGS to detect multidrug resistance in culture-negative biopsies and on clinical specimens collected during the standard of care [39].

Studies show that clarithromycin resistance is based on point mutations at nucleotide positions A2146 and A2147 in the 23S rRNA gene [22,39]. The rRNA 16S gene is a much more sensitive method for detecting *H. pylori* in gastric biopsies compared to other methods [22].

Sequencing *H. pylori* DNA from gastric biopsy specimens is a laborious method. *H. pylori* must be cultured from multiple gastric biopsy specimens, then, multiple colonies must be picked from agar plates for DNA extraction in order not to miss the drug-resistant subpopulations; the strains should be sequenced with sufficient coverage to detect heteroresistance; usually, multiple susceptible and resistant strains of *H. pylori* are sequenced [26].

The detection of *H. pylori* DNA in stool samples is a very convenient, fast, sensitive, and accurate method. Stool RT-PCR analysis can detect *H. pylori* DNA sequences and antibiotic resistance point mutations. The conducted meta-analysis showed that most diagnostic candidate genes identified in stool samples were 23S rRNA, 16S rRNA, and glmM. Stool DNA PCR had a performance of 71% (95% CI: 68–73) sensitivity and 96% (95% CI: 94–97) specificity in the diagnosis of *H. pylori*. Analysis showed that the 23S rRNA gene has high sensitivity for the detection of *H. pylori* in clinical samples [40]. Three mutations (A2142G, A2143G, and A2142C) in a gene in 23S rRNA were associated with *H. pylori* resistance to clarithromycin, and these mutations have been associated with treatment failure [22].

Undoubtedly, stool DNA PCR has its advantages: it gives faster results, fewer bacteria are required in the sample for analysis, it does not need special processing supplies or transportation of the material, and the result can be obtained in a fairly short time (<4 h).

Despite the high specificity of the test, a number of studies have revealed a high percentage of false-positive results, especially when the test is carried out 4–6 weeks after successful eradication therapy. False-positive results in treated patients can be explained by persistence in the feces of coccoidal forms of *H. pylori*, which, over time, begin to decrease and completely disappear at 8–12 weeks [41].

In geographic regions with high clarithromycin resistance, stool RT-PCR testing with determination of clarithromycin resistance is a useful diagnostic option for young dyspeptic patients who do not require endoscopy and should preferably be treated with clarithromycin-containing regimens [42].

## 5. Conclusions

The high prevalence and etiopathogenetic relationship of *H. pylori* with the most significant diseases of the stomach highlights the need to optimize the diagnosis of this infection, taking into account the sensitivity and specificity of the tests, as well as the conditions for their use. The infection must be detected before therapy is prescribed, and its success must be confirmed after treatment.

The developments of current diagnostic methods allow for a more accurate and reliable diagnosis of *H. pylori* infection. The choice of method will depend on the accessibility, their advantages and disadvantages, sensitivity and specificity, and different clinical circumstances of each patient.

Leading international experts dictate the rules for the diagnosis of *H. pylori* infection; however, the majority of mistakes are still made when assessing the effectiveness of eradication, namely, the use of inadequate methods or lack of control. According to the European Registry on *H. pylori* management (Hp-EuReg), confirmation of the eradication was performed in 94% of the cases [43].

## Figures and Tables

**Table 1 diagnostics-11-01458-t001:** Overview of the diagnostic methods for *H. pylori*.

	Initial Diagnosis	Follow-up after Eradication	Requires Excluding PPI, Antibiotics, Bismuth Before Testing	Gastroduodenal Bleeding	Detection of Antibiotic Resistance	Sensitivity	Specificity
Invasive (require upper endoscopy)							
Histology	+	+	+	−	−	91–93%	100%
RUT	+	−	+	−	−	85–95%	95–100%
Culture	+	−	+	−	+	76–90%	100%
Molecular method (PCR)	+	+	+	+	+	95%	95%
Non-Invasive							
UBT	+	+	+	+	−	96–100%	93–100%
SAT	+	+	+	−	−	95.5%	97.6%
Serology	+	−	−	+	−	76–84%	79–90%
Stool PCR test	+	−	+	+	+	71%	96%

## Data Availability

Not applicable.

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
