# Peer review of "Current *Helicobacter pylori* Diagnostics"

_diagnostics, 2021, doi:10.3390/diagnostics11081458_

Round 1
Reviewer 1 Report
The review by Dmitry S. Bordin et al. describe current H.pylori diagnostics.
This review include numerous point that have to be revised and/or completed in order to be published.
Introduction :
Line 52 to 55 is not true for all methods and so have to be discussed. The table 2 could be included in this part of the manuscript to discuss these point more easily.
Table 1 and 2 could be fuse in one table including approximated price per sample.
Line 49 (and other iterations...) Molecular biology in feces is not correctly referenced. Prefer to cite the original article than another review. for example Pichon et al. J.Clin.Microbiol 2020. PMID: 31996442
Line 60 : cost-effectiveness is very difficult to state. Give the approximate cost per sample for all described techniques.
Part 2 Invasive Methods for H.pylori diagnostics
Global : italicize "vs."; "et al."; "e.g." - number <12 have to be written in full letters
Line 84 : what was the gold standard in this study?
Line 91-92 : give reference
Line 95 : "expensive and time-consuming method" give details.
Line 97 : a subpart has to be indicated more appropriately
Part 2.2 : Histology is an important part of the justification of the invasiveness needing. Nevertheless, I think that the authors have te give more details about the heterogeneity of the Hp repartition in an host (see Pichon et al. J.Clin.Med 2020 PMID: 32878081. Maybe in part 2.4?
Justify the need for stopping antibiotics and PPI 2-4 weeks before for histology examination.
Line 191 : a problem occurred in reference addition ?
Line 194 : describe more appropriately the assays (manufacturers' name, city, country).
Line 202 : justify why these clinical outcome impact the results of the assays
Part 2.4 : authors have to give more details about "optimal transporting" and "optimal condition " (line 218 and 224)
Line 222 : which antibiotics are currently recommended to test?
Part 3 Non- invasive Methods for H.pylori diagnostics
A typo is present "non-" invasive methods I suppose.
Could the authors give more details about the analytical difference between 14C and 13C UBT?
Replace "minus 20 degrees" by "-20°C"
Line 326 : what are cross-reactive antibodies targeting?
Line 332 : what is "low prevalence" threshold ?
Line 337 : as these results are conflicting, it could be interesting to give more details.
Part 4 Helicobacter Pylori diagnostic data register.
I think that this part is not of high interest in this manuscript and is present at a wrong place. Please suppress or modify the manuscript to put it between part 1 and 2?
Website could be referenced as other bibliographic ressources.
Line 439-449 is only justified by the Russian nationality of the authors. Please suppress.
Line 406 to 409 (including table 2 have to be included in the introduction.Author Response
We have made changes accordingly with the reviewer’s comments

Reviewer 2 Report
The review article is written correctly. While the subject matter is not novel, information on new diagnostic techniques has been introduced, which has a positive effect on the reception and usability of the manuscript. I have some comments being presented below. I strongly encourage the Authors to read them and make appropriate corrections.
The list of suggestions:
- “caused by this infection…” -> caused by this pathogen [line 10]
- “diagnostic tests for H. pylori detection” -> diagnostic tests for H. pylori detection [line 15]
- “molecular testing from stool or a molecular methods-PCR study with determination of H. pylori DNA in feces” -> what is a difference here between these two? [lines 48-49]
- Table 1: I belive that the expression should be homogenous to not introduce misunderstandings. It should be “PCR using gastric biopsy specimens” and “PCR from stool” OR “molecular testing of gastric biopsy specimens” and “molecular testing of stool”
- “the best option is 2 biopsies from the antrum” -> the best option is two biopsies from the antrum [lines 128-129]
- “high false-negative rate [Error! Bookmark not defined.]” -> please a proper reference here [line 191]
- “urease-producing bacteria such as Staphylococcus capitis urealiticum” ->urease-producing bacteria such as Staphylococcus capitis subsp. ureolyticus [line 203]
- “urease-producing bacteria such as Staphylococcus capitis urealiticum” -> only this? What with Streptococcus spp. and others? Please add a short list of urease-producing bacteria able to colonize the stomach [line 203]
- “Maastricht IV Consensus Report recommends …” -> I believe that it should be: Maastricht V Consensus Report recommends [line 245]
- “3. Invasive Methods for H. pylori Diagnostics” -> 3. Non-Invasive Methods for H. pylori Diagnostics [line 250]
- “Antibodies to immunoglobulins …” -> what is the meaning of this? Maybe: Immunoglobulins against antigens … [line 327]
- In the “3.4 Molecular methods” section there is a lack of logic presented in the Table 1. Why the molecular testing of gastric biopsy specimens is presented here (in the section of non-invasive methods)? Please correct.
- Table 2 should include references (there are very precise values of both sensitivity and specificity)
- I strongly believe that the section “4. Helicobacter pylori diagnosis data register” should be removed because the results refer only to Russia and are therefore are of little relevance in the context of the current review (data cannot be extrapolated to the whole world). These results, if unpublished, should be presented independently in the form of a research paper or a brief report.
Author Response
We have made changes accordingly with the reviewer’s comments

Round 2
Reviewer 1 Report
It seems that most of my review#1 has ben adressed but my reviewing#2 would need a point to point response to my preivous comments. Please perform.
Waiting for it, I maintain my classification.
Author Response
Dear REVIEWER, thank you for kind review. We have made changes with regard of your recommendations.

Reviewer 2 Report
The Authors of the article responded to the most of suggested amendments. Thus, I believe that the quality of the manuscript has improved.
Author Response

(The authors gave the same response as above.)
